

# Sky radiance at a coastline and effects of land and ocean reflectivities

Axel Kreuter[1,2], Mario Blumthaler[1], Martin Tiefengraber[2,3], Richard Kift[4], and Ann R. Webb[4]

[1]Division for Biomedical Physics, Innsbruck Medical University, Innsbruck, Austria
[2]LuftBlick, Earth observation technologies, Mutters, Austria
[3]Institute for Meteorology and Geophysics, University of Innsbruck, Innsbruck, Austria
[4]School of Earth Atmospheric and Environmental Sciences, University of Manchester, Manchester, UK

*Correspondence to*: Axel Kreuter (axel.kreuter@gmail.com)

**Abstract.** We present a unique case study of the spectral sky radiance distribution above a coastline. Results are shown from a measurement campaign in Italy involving three diode array spectroradiometers which are compared to 3-D model simulations from the Monte Carlo model MYSTIC. On the coast, the surrounding is split into two regions, a diffusely reflecting land surface and a water surface which features a highly anisotropic reflectance function. The reflectivities and hence the resulting radiances are a nontrivial function of solar zenith and azimuth angle and wavelength. We show that for low solar zenith angles (SZA) around noon, the higher land albedo causes the sky radiance at 20° above the horizon to increase by 50% in the near infrared at 850 nm for viewing directions towards the land with respect to the ocean. Comparing morning and afternoon radiances highlights the effect of the ocean's sun glint at high SZA which contributes around 10% to the measured radiance ratios. The model simulations generally agree with the measurements to better than 10%. We investigate the individual effects of model input parameters representing land and ocean albedo and aerosols. Different land and ocean BRDFs do not generally improve the model agreement. However, consideration of the uncertainties in the diurnal variation of aerosol optical depth can explain the remaining discrepancies between measurements and model. We further investigate the anisotropy effect of the ocean BRDF which is featured in the zenith radiances. Again, the uncertainty of the aerosol loading is dominant and obscures the modelled sun glint effect of 7% at 650 nm. Finally, we show that the effect on the zenith radiance is restricted to a few kilometres from the coast line by model simulations along a perpendicular transect and by comparing the radiances at the coast to those measured at a site 15 km inland. Our findings are relevant to, for example, ground based remote sensing of aerosol characteristics since a common technique is based on sky radiance measurements along the solar almucantar.

## 1 Introduction

In the absence of clouds, the solar radiation in the UV- NIR spectral region is scattered by air molecules and aerosols which renders the radiant, blue sky. Part of the down welling radiation is reflected back from the earth's surface and contributes to the radiance (Kylling and Mayer, 2001). Gases may also absorb the light on its path through the atmosphere and characteristically modify the spectrum.

The spectral sky radiance thus carries information about the atmospheric composition, trace gases, aerosols and the underlying surface and is the basis for remote sensing of the atmosphere. For example, differential optical absorption spectroscopy (DOAS)(Platt and Stutz, 2008) utilizes the relative radiances of the spectrum to



determine trace gas concentrations. The absolute radiance is analysed, e.g. for the determination of the microphysical properties of aerosols (Dubovik and King, 2000).

Typically, land reflectance is diffuse, that is isotropic and independent of viewing angle, and is described by
Lambert's cosine law. Integration of the bi-directional reflectance function (BRDF) over all viewing angles is proportional to the albedo (Coakley, 2003). In the UV – VIS spectral range, the Lambertian albedo for land surfaces ranges from almost 1 for fresh snow to around 0.2 for forest and 0.05 for water. In the NIR, vegetation has a higher reflectivity of about 0.4 (Coakley, 2003). The spectral albedo features are useful for satellite remote sensing of e.g. vegetation index and land cover (Hansen et al., 2000).

Compared to land, the ocean surface has a contrasting reflectivity property because it can be highly anisotropic and directional so the BRDF is strongly peaked at a certain reflection angle (Cox and Munk, 1954). The BRDF depends strongly on wind speed: A calm ocean shows a highly specular reflection, a phenomenon known as sun glint which is best observed at high solar zenith angles (SZA). At higher wind speeds the water surface is ruffled
and the reflection becomes more isotropic and tends towards a Lambertian surface. Satellite observations of the sun glint allow a good estimate of the surface wind speed above the oceans (Wald and Monget, 1983).

Besides a weak wavelength dependence of the water's index of refraction which determines the reflectivity of a plane water surface by Fresnel's equation (e.g. Hecht, 2002), the wavelength dependence of the sun glint is more
implicit. A specular reflection is only effective for a directional light source. The strong increase of Rayleigh scattering efficiency with shorter wavelengths causes a large diffuse component of the irradiance in the UV, i.e. a small direct to diffuse ratio. For VIS and NIR wavelengths, the direct component is much larger and hence the specular reflection.

In addition, the direct to diffuse ratio of the global irradiance generally decreases with higher solar zenith angles
and the light path through the atmosphere lengthens. The resulting reflection property of the ocean has an intricate dependence on both wavelength and SZA.

Quantifying the effect of an inhomogeneous albedo distribution on the solar irradiance and sky radiance has been recognized as a challenging problem, since it requires the use of a three dimensional radiative transfer model. As
an extreme example, the arctic regions are characterized by highly inhomogeneous albedo distributions due to the contrast of highly reflective snow (enhancing global irradiance by up to 50% (e.g. Blumthaler, 2007)) and dark ocean in the UV-VIS which has been subject of several experimental and model studies in the past (Kreuter et al., 2014, Ricchiazzi et al., 2002, Deguenther et al.,1998) .

In this paper, we report on spectral sky radiance measurements in the VIS-NIR range at a coastline which partitions the surface into two opposing segments: land and ocean. With detailed 3D model simulations, we investigate this radiative transfer problem theoretically in pursuit of a deeper understanding of the components of the sky radiance above such an intriguingly complex surface configuration. Following the conventional structure, we describe our methods before we present our measurement and model results and discuss relevant associated
aspects.



## 2 Methods

### 2.1 Measurements

A dedicated measurement campaign was performed for two weeks in September 2015 in Grottammare south of Ancona on the Adriatic coast in Italy. This site was chosen because it features a fairly straight coast line that runs close to the north-south direction. In this case, the course of the solar azimuth during the day is symmetric to the coast, which is ideal for studying the anisotropy of the ocean. We also favored a location with low wind (wave) conditions and flat topography of the land. The beginning of autumn offered a high chance of cloud free skies.

The measurement instruments included three diode array spectroradiometers designed to measure global irradiances and radiances in the UV-VIS-NIR spectral range. Two diode array spectroradiometers had global input optics which were fitted with a shadow tube to measure diffuse irradiance from the zenith (because surface reflectivities affect only the diffuse radiance and hence the effect is decreased in the global irradiance by the direct sun component). The shadow tube was a circular tube made of aluminum, coated black on the inside and mounted on top of the global optics. The tube shadowed the sky at elevation angles less than 60° which ensured that the sun was always occluded. The maximum solar elevation at the time of the campaign was about 51°. These measurements can also be regarded as measurements of the zenith radiance with a 60° field of view. Both instruments, in the following referred to as DA1 and DA2, recorded synchronous spectra every 10 min. The instruments have been well characterized in the laboratory and during previous campaigns which is described in more detail (Kreuter et al., 2014, Kreuter and Blumthaler, 2009).

The third diode array spectroradiometer was the Pandora-2s instrument (PAN). It has an input optics with 2.5° field of view mounted on a 2-axis tracker to measure direct sun and sky radiances. It was specifically designed for the retrieval of trace gases and aerosols and has been well characterized (Herman et al., 2009). The spectral range of the Pandora covers the UV-VIS-NIR range 300-900 nm. For this campaign, it was programmed to measure the spectral radiance for a set of azimuth angles at 70° zenith angle. The measurement duration for a complete scan was 4 minutes and the timestamp was assigned to the time at the middle of measurement. The order of the angles was reversed in the afternoon, so that the sequences were symmetric around solar noon. The scans were scheduled every 30 min. In between, we also performed direct sun measurements and radiance scans along the principal plane which includes the zenith.

We maintained two measurement sites which are shown in Fig. 1: One directly on the coast, 100 m from the water line, and one 15 km inland. Two instruments (DA1 and PAN) were located at the coastal site, while the other instrument DA2 was set up at the inland site. The instruments were calibrated for global irradiance and radiance, respectively. However, since we are considering relative ratios here, the absolute calibration is irrelevant and only relative radiometric instrument stability has to be ensured, by temperature-stabilizing the instruments. Before the field measurements, DA1 and DA2 were operated together at the coastal site to check instrument stability.

Two auxiliary instruments were deployed at the coastal site to complement the observations: A sun photometer to measure the aerosol optical depth (AOD) at four wavelength channels, and an all-sky camera to monitor the



sky and verify cloud free conditions. After two weeks of measurements, one full day (12.9.2015) and one half day (10.9.2015) were completely cloud free at both sites and hence suitable for this investigation.

## 2.2 Model simulations

For modelling the sky radiances, we apply the 3D Monte Carlo radiative transfer model MYSTIC within the libRadtran package (Mayer, 2009, Mayer 2010) in a cooperation agreement with the model developers. LibRadtran is a freely available, open source project, however currently, only a 1D version of MYSTIC is included in the public distribution (Mayer and Kylling, 2005).

In backward mode, MYSTIC randomly traces photons originating from the detector through the atmosphere. At each scatter or surface reflection event, a local estimate is performed, i.e. the probability that the photon scatters/reflects toward the sun and reaches the sun without being extinct is calculated. The sum over all local estimates, divided by the number of simulated photons, gives the transmittance weighted by the cosine of the SZA. The spectral radiances are computed for the two measurement locations for the relevant day, in accordance

with the solar geometry of the measurement schedule. Each simulated radiance is a result of $>10^5$ sampled photons ensuring a statistical error of <1% standard deviation.

For the modelled standard scenario, the atmosphere is assumed cloud free with a standard midlatitude summer AFGL vertical profile (Anderson et al., 1986). The atmosphere has a plane-parallel geometry since the spherical

atmosphere has not yet been implemented in MYSTIC in combination with a 2D surface. This approximation is usually well justified where only SZA<80° are considered, especially for ratios of radiances at identical SZA. Aerosol properties are specified according to the OPAC aerosol type continental average (Hess et al., 1998), with Angstrom alpha and beta parameters scaled to 1.4 and 0.05, respectively.

The 2D surface is specified by a 400 x 400 km$^2$ grid with a 10 km resolution. The elevation at each grid point has been set constant to 100 m for land and 0 m for ocean. The coast is a straight line at an angle 20° (anticlockwise) from the north-south direction, depicted in Fig. 1. Each grid element also has a surface reflection property given by the bidirectional reflectance distribution function (BRDF). Here, we consider two BRDFs that for water (ocean) and that for land.


Water reflection is modelled applying the commonly used Cox and Munk (CM) parameterized BRDF function of wind speed and direction (Cox and Munk, 1954). The wind speed has been set to 5 m/s, estimated from the visual appearance of the ocean and local wind measurements. The land is modelled as a Lambertian surface with an albedo of 0.05, 0.18 and 0.3 for 450, 650 and 850 nm respectively. This is supported by the albedo products

from the MODIS sensors on-board Aqua and Terra satellites (Schaaf et al., 2011). The mean albedo retrieved from band 2 (841 nm – 876 nm) for the surrounding during September 2015 has been determined as 0.32 and 0.29 for the white-sky and black-sky albedo, respectively (referring to the albedo with and without scattering).

We note that we explicitly model a simplified scenario in terms of geography here to illustrate the general

features of the sky radiance at a coastline with respect to land and ocean reflectivities. In a model study on the





global irradiances in Svalbard (Kreuter et al., 2014) we found that the effect of topography (which was comparable to the case here) could be neglected.

## 3 Results

### 3.1 Sky radiance at 70° viewing zenith angle

First, we look at the azimuthal scans of the sky radiance at 850 nm from the Pandora instrument located at the coastal site and investigate its dependence on the solar azimuth. The azimuthal scans were performed at 70° viewing zenith angle, over the course of the day with varying SZA. The comparison of the radiance in the morning and afternoon at the same SZA are shown in Fig. 2. SZAs of 78.2, 67.5, 57.4 and 41.9, corresponding to measurement times of 5.2, 4.2, 3.2 and 1.2 hours from local noon, respectively, will be used as representative
solar positions throughout this study.

Left and right panels show measured and modeled radiances, respectively. The viewing azimuth angle is defined relative to the sun at 0° and counted clockwise. The ocean–land distribution (blue-green patches), the position of the instrument at the coast (small black cross), the azimuth of the sun (dark blobs) and the measured azimuth
angles (small white ticks, 90°, 180°, and 270° are marked black) are indicated in the sketches on the right .

In general, the radiance increases towards the direction of the sun (viewing azimuth angle <90° and >270°) which is mainly due to the forward scattering of aerosols. At 180°, in viewing direction opposing the sun, the sky radiance has a local maximum which decreases with decreasing SZA. This maximum is associated with
molecular backscattering. Furthermore, the sky radiance at angles around 180° shows the largest difference between am and pm. As viewed from the coast in the morning, the sun is over the ocean and at 180° azimuth angle, we are looking towards the horizon of the land which has a higher albedo which increases the radiance. The qualitative features are well reproduced in the model. Only at 78° SZA the difference between the modeled radiances of am and pm is notably smaller than observed.


For NIR wavelengths, the coastline divides the surface around the observer at the coast into a high (land) and a low (ocean) albedo. Since the coastline is almost north/south, this breaks the symmetry in the sky radiance azimuthal scans in two ways: In an asymmetry between the left and the right hemisphere above the coastline and in an asymmetry with respect to the solar azimuth i.e. between morning and afternoon. For shorter wavelengths
towards the blue range of the spectrum, land and ocean reflectivities are similar and the anisotropy is expected to disappear. In the following, we will investigate the respective ratios which highlight this asymmetry of the radiance above the albedo distribution.

### 3.2 Ratios of the sky radiance between the right and the left sky hemisphere

Now we consider the ratios of the radiance in the right and left hemisphere, i.e. the symmetry of the radiance
with respect to the principal plane (the plane through the observer, the zenith and the sun). The ratios at 70° viewing zenith angle for the four selected SZA in the morning and afternoon are shown in Fig.3. The viewing azimuth angles from 50° to 175° are relative to the principal plane. As above, the right panel of Fig.3 shows the orientation of the radiance frame of reference with respect to the coastline.



As a first general observation, the ratios have a maximum between 90° and 135° viewing azimuth and increase with decreasing SZA. In the morning and afternoon, at high SZA, the ocean land distribution is almost symmetric with respect to the principal plane, resulting in ratios <10%. At 67.5° SZA in the afternoon, the principal plane is perpendicular to the coast line and the ratio is unity. With decreasing SZA towards noon, the relative frame of reference for the azimuth rotates and the right-left ratios increase. At 41.9° SZA in the morning, the principal plane is aligned with the coastline maximizing the albedo asymmetry and hence the right-left ratio.

The maximum of the ratios is around 1.4 for 850 nm, i.e. at 70° zenith angle and 120° azimuth angle, the sky in the NIR is 40% brighter over land than over the ocean. The ratios decreases with wavelength and vanish for a wavelength of 450 nm where both land and ocean albedo are about 0.05. Considering the model simulations, we note that the characteristic features of the ratios with respect to azimuth angle, SZA and wavelength are well reproduced. Quantitatively, we generally have an agreement of measurements and model simulations of better than 5%. In particular, the good agreement of the ratios at 850 nm at 41.9° SZA, which have the highest sensitivity to the albedo distribution, indicate that the Lambertian albedo model for the land with albedo 0.3 is appropriate.

### 3.3 Ratios of the sky radiance between morning and afternoon

Next we will examine the asymmetry of the radiances between morning and afternoon. In Fig. 4, we show measured and modeled ratios of the radiances between morning (am) and afternoon (pm) at 70° viewing zenith angle at three wavelengths at the coastal site at four SZAs. The viewing azimuth angle is relative to the solar azimuth as in Fig.2.

The ratios are close to unity for viewing azimuth angles towards the direction of the sun (azimuths <50° and >270°) and maximal around 180° from the sun. For short wavelengths, the ratios are always close to one, while the maxima increase with SZA and wavelength. The widths of the maxima decrease with decreasing SZA which can be understood by looking at the difference of the albedo between am and pm for each viewing azimuth.

The general features of the ratios at each SZA are well reproduced by the model simulations. However, the magnitude of the modeled ratios is systematically smaller by about 10%. As a next step, we revisit the model input parameters and assess a plausible uncertainty for each relevant parameter and hence estimate the resulting model uncertainties.

### 3.4 Model sensitivity study and discussion

The most relevant input parameters that affect our study are the land and ocean surface reflection properties as well as aerosol loading. In the following we will quantify the respective model sensitivities to these three parameters.

First, we have assumed an idealized Lambertian model for the land reflectivity which may not be perfectly valid in reality. Second, the ocean's BRDF model depends on wind speed which was estimated from personal observations and from the visual appearance of the sun glint. There is an uncertainty from that as well as perhaps





the parameterization of the model itself. Third, the AOD measurements from both the sun photometer and the Pandora instrument indicate a constant AOD (beta=0.05) with a remaining uncertainty from the standard deviation of about 0.005. Focusing on these factors individually, we set up three alternative model scenarios with modified model input parameters described in Table 1. All other parameters are the same as in the standard

scenario.

The 'land' scenario involves modeling the land reflectivity with a slightly anisotropic BRDF applying the semi empirical parameterization by Rahman, Pinty and Verstraete (RPV) (Rahman et al., 1993). The RPV model includes three parameters ($\rho0$, k and $\Theta$) to describe a generalized surface reflection function. Here we use the

parameters which have been given for a pasture type surface which could in principle apply to much of the surrounding of the measurement sites. The pasture land BRDF is close to a Lambertian albedo and differs essentially by an increased reflection in the backscatter direction. This so-called 'hot-spot' is often noticeable at high SZA when the sun is in the back of the observer which is a result from geometric shading on structured surfaces. The effect is opposite to that of a specular reflection, albeit much weaker.

In the 'ocean' scenario, we modify the ocean BRDF, by increasing the wind speed in the CM parameterization. This is expected to increase the isotropy of the BRDF and we confirm that above 30 m/s the ratios are (within noise) equal to the ratios modeled with a Lambertian albedo 0.05 which is shown here as the limiting, isotropic case. In the 'aerosol' scenario, we change the AOD (the Angstrom beta parameter) by its uncertainty of 0.005.

We model the radiances with a beta of 0.055 in the morning and 0.045 in the afternoon and vice versa, keeping the Angstrom exponent alpha constant.

For these scenarios, the resulting right-left ratios as well as the am-pm ratios for 850 nm at the four selected SZA are shown in Fig. 5. This simultaneous comparison of both ratios with model and measurements at different SZA

allows an intricate assessment of the model sensitivity to the various parameters.

Inspecting the first scenario with a modified land BRDF for pasture land, it is noticeable that both right-left and am-pm ratios are generally increased. Right-left ratios overestimate the measurements by up to 20%. Since the land and ocean are in opposite directions from the observer at the coast, the hotspot has a similar effect on the

radiance as the sun glint. The am-pm ratio increases by 5% at 78.2° SZA and improves the agreement with the measurement, which is also the case for 41.9° SZA. At 67.5° SZA the am-pm ratio increases to 1.5 which then overestimates the measurement by almost 15%.

When the ocean is modeled as a Lambertian reflector with albedo 0.05, the right-left ratios are mainly

unaffected, less than 5% at 41.9° SZA, compared to the standard scenario of the CM parameterization. The am-pm ratio is reduced by just over 10% for all azimuth angles at 78.2° SZA. The amplitude of the angular dependence is not affected. The difference quantifies the contribution of the ocean's BRDF, the sun glint. This effect decreases with decreasing SZA to below 5% at 42° SZA. This is plausible since the sun glint is most prominent at high SZA as one can expect more specular reflection off a flat water surface at glancing angles of

incident purely from Fresnel's law. The model-measurement agreement is reduced particularly at 78.2° SZA except for 50° azimuth angle. We note that the Cox and Munk model for the ocean BRDF is not strictly suited



for SZA>80°, so its validity may already be limited for 78° SZA. Similarly, the validity of the land albedo, whether Lambertian or RPV BRDF, might not hold perfectly for higher SZA.

In the third scenario, we look at the sensitivity of the ratios to AOD changes during the day. The right-left ratios are not affected for this scenario because they are evaluated at one point in time (the measurement time is negligible here). For the am-pm ratios, small diurnal variations of 0.01 of the AOD result in an increase or decrease of the ratios of 10% depending on whether the AOD was 0.045 in the morning and 0.055 in the afternoon or vice versa. The corresponding uncertainty is depicted as the grey band in Fig. 5. The variation is slightly bigger for viewing azimuth angles towards the sun which is caused by the prominent forwardly weighted scattering of aerosols. In simulations with other types of aerosols (OPAC type urban and marine aerosols with different properties like SSA and phase function) we found a negligible effect on the ratios. In this context, additional uncertainties could also be caused by an inhomogeneous distribution of aerosols or even thin clouds far away from the observer.

Combining these observations, each scenario improves the model agreement at least for some viewing angles or SZA, but none of them constitutes a convincing universal improvement in a way that improves the model for all viewing azimuth angles and SZA. However, the sensitivity of the ratios is selective regarding model scenario, SZA and viewing angle. For example, the right-left ratios at 41.9° SZA are mostly sensitive to the land albedo or BRDF because the sun glint effect is small and the ratio is not affected by the AOD uncertainty. A good agreement of the standard scenario indicates an appropriate albedo model especially for low SZA, although the pasture land BRDF scenario improves the model agreement for high SZA for the am-pm ratios. For high SZA, the am-pm ratios are sensitive to the ocean BRDF and the 'ocean' scenario shows that the CM model for the ocean BRDF with higher wind speeds reduces the agreement with the measurements.

Considering the uncertainty band due to the AOD uncertainty, the discrepancies of the modeled and measured ratios are well explained. A further refinement of the land and ocean reflectivity models, however, is not feasible within this study due to this uncertainty.

### 3.5 Zenith radiance on the coast

Finally, we investigate the zenith radiance measured with the shadow tubes in order to gain another perspective of the separate effects of BRDF anisotropy and Lambertian (or effective) albedo differences between land and ocean. The zenith radiance only depends on the effective albedo and not on the distribution. Above a Lambertian surface the zenith radiance is independent of the solar azimuth and independent on the albedo distribution. However, if part of the surface has a non Lambertian reflection property, e.g. shows a specular reflection like the ocean's sun glint at low solar elevation, then the zenith radiance would not be invariant to the solar position.

Both solar azimuth angle and SZA have an implicit importance here. On the one hand the ocean BRDF is strongly dependent on SZA, i.e. the higher the SZA, the more pronounced the sun glint. On the other hand, from an observer's point of view at the coast, it makes a difference whether the sun is over the ocean or whether it is over the land, i.e. whether the specular reflected photons are reflected towards the observer's zenith or not.





To investigate this asymmetry of the zenith radiance with respect to the solar position, we consider the ratios of the radiance at identical SZA in the morning and afternoon, respectively. Measured and modeled am-pm zenith radiances for 450 nm and 650 nm at the coastal site are shown in Fig. 6. The NIR wavelength of 850 nm is not included in the analysis here because of a spatial stray light problem. The tubular shadow band of instrument

DA1 was covered with black felt on the inside, which is only absorptive for visible wavelengths while being reflective for the NIR part of the spectrum and may perturb the signal by a reflection of the direct sun.

First, the ratios of instrument DA1 for a 60° FOV are in good agreement with those of the Pandora instrument for a 2.5° FOV zenith radiance (open circles). The measured am-pm ratios seem to be dominated by a large

variation compared to the systematic increase with SZA of the ratios modeled with a constant AOD. Without any atmospheric disturbances, the modeled zenith radiance in the morning is 7% higher than in the evening for large SZA above 70° for 650 nm and 4% at 450 nm. This dependence is characteristic for the ocean's sun glint.

The SZA dependence of the sun glint has been explained before, however the wavelength dependence is less

trivial. A specular reflective surface is, of course, only relevant for the direct sun (or a directional light source in general). At longer wavelengths the ratio of the direct to diffuse irradiance is higher because of less scattering by air molecules and aerosols. On the other hand, a lower scattering probability also reduces the probability that the reflected light from the surface is scattered back towards the observer. So we have to consider two opposing effects, which are difficult to balance against each other from these basic arguments. The model has shown that

the sun glint effect is indeed more pronounced at longer wavelengths.
The measured ratios are clearly dominated by AOD variation during the day but are within the range of modeled ratios with a variation of β between 0.045 and 0.055 (grey band). So the zenith radiance is highly sensitive to scattering by aerosols and even a small change of 0.01 in β, the range of the typical measurement uncertainty of well calibrated sun photometers, causes a change of the radiance of up to 15% and would obscure the effect of

ocean BRDF for one wavelength. However the relative difference between the two wavelengths, increasing from about zero at 60° SZA up to 5% at 75° SZA, remains as the characteristic signal of the sun glint.

### 3.6 Comparison of the zenith radiance between the coastal and inland site

Furthermore, we want to investigate the translational dimension of the radiance above a non-uniform albedo distribution and we look at the zenith radiance with respect to the position of the observer relative to the

coastline.

The dependence of the zenith radiance on the solar azimuth, i.e. the effect from the ocean BRDF discussed above cannot be expected when the observer is in the midst of the ocean. Although the sky radiance distribution is strongly affected by the ocean specular reflection, the zenith radiance should be identical, morning and

afternoon, since the geometry is rotationally invariant about the zenith. Of course, the same holds when the observer is surrounded by land surface. So the am-pm ratios should have a maximum at the coastline and decrease to unity after moving some distance either inland or towards the ocean. Assuming a Lambertian land albedo, this maximum is caused only by the ocean's strongly anisotropic BRDF.





In Fig. 7 we show the am-pm ratios of the zenith radiance measured simultaneously at the two measurement sites, at the coast and 15 km inland, respectively. Also shown are the modeled am-pm ratios of each site, as a function of the SZA for two wavelengths, 450 nm and 650 nm. Measured simultaneous ratios are only applicable for a single day in the SZA range 65° - 75°. Otherwise high clouds to the west of the inland site or convective
clouds around noon spoiled the measurements. The am-pm ratios at the coast increase with SZA and wavelength while little variation can be identified for the ratios measured 15 km inland.

Although the measured values show a considerable variability (due to a AOD variability as discussed above) two small but important features are reproduced by the model: the am-pm ratios are higher at the coast and increase
with wavelength. Especially, the increasing difference of the ratios between the two wavelengths again demonstrates the effect of the ocean's sun glint.

The spatial extent of the sun glint effect is shown in the right panel of Fig. 7: simulated am-pm ratios along a transect perpendicular to the coast line at 70° SZA. The am-pm ratios are maximal directly at the coast and
reduce to unity over the ocean and over the land about 30 km from the coast.

## 4 Conclusions

We have measured and modeled sky radiances at a coastline, where a near-Lambertian land albedo contrasts with the highly anisotropic ocean BRDF. At short wavelengths around 450 nm, the effective albedos of both surfaces are similar and low, while towards the NIR spectral range the albedo of the land is significantly higher.
We have looked at various ratios between radiances at different solar azimuth angles for specific wavelengths and SZA to investigate the effects of this albedo distribution.

First, we investigated the asymmetry of the sky radiance at the coastal site by comparing the radiances in the right and the left viewing directions (relative to the sun) at 70° zenith angle. We have shown that the radiance in
viewing direction towards the land can be up to 50% higher compared to viewing direction towards the ocean at 850 nm and SZA of 40°. This is the effect of the higher land albedo compared to the ocean.

The same is also apparent in the ratio between the radiances in the morning and afternoon, especially towards noon for viewing directions opposing the sun. The am-pm ratios decrease with increasing SZA and are about 1.3
for 850 nm at 78.2° SZA. For these ratios, the anisotropic ocean BRDF, the sun glint, also plays a role, especially at high SZA. The discrepancies between measurements and 3D model simulations are generally below 10%.

A sensitivity study shows how the ratios are affected by model input parameters regarding land and ocean BRDF
and a diurnal variation of aerosol loading. The right-left ratios which are independent of AOD variations during the day indicate that a Lambertian land albedo of 0.3 in the NIR is appropriate for low SZA although a slight anisotropy of the land BRDF improves the model agreement at 78.2° and 41.9° SZA in the am-pm ratios. The contribution of the sun glint to the asymmetry between morning and afternoon radiances is about 10% at high



SZA. The high sensitivity to slight changes in the AOD indicates an uncertainty range for the model simulations that results in a satisfactory overall agreement with the measurements.

In order to only focus on the anisotropic part of the reflectivity, mainly the ocean BRDF, we investigate the zenith radiance. For a constant AOD during the day, the modeled am-pm ratio of the zenith radiance has a maximum of 7% for 650 nm and 4% at 450 nm for SZA > 70°. However, due to sensitivity of the zenith radiance to aerosol scattering, the ratios are easily dominated by diurnal AOD variations, which was the case for our measurements. A varying Angstrom beta parameter between 0.045 and 0.055 causes changes in the ratio by up to 15%. Nevertheless, the difference between 450 nm and 650 nm of up to 5% at 75° SZA remains as a characteristic of the sun glint. It was further shown that this effect on the zenith radiance will be negligible further than 30 km away from the coast.

While the zenith radiance is weakly affected by the inhomogeneous surface reflectance distribution, the radiance at higher viewing zenith angles (as shown here for 70°) may be modified significantly. At a typical coastline, the radiance can differ up to 50% in the NIR spectral range compared to simplified 1D radiative transfer model simulations for the specific conditions described here. For the geometry of satellite observations, the radiances within a few kilometers from the coast can be increased by 10% due to the sun glint, however the measured reflectance signals will typically be dominated by uncertainties of ground albedo and aerosols.

Our results are relevant for any ground based remote sensing of radiances near the coast aiming to retrieve atmospheric components. For example, within AERONET, radiance measurements in the almucantar from sun photometers are used to retrieve aerosol microphysical properties, such as the size distribution and the index of refraction.

**Data availability**

For data access please contact the corresponding author.

**Competing interests**

The authors declare that they have no conflict of interest.

**Acknowledgements**

The authors would like to thank R. Buras, C. Emde and B. Mayer for the help and support for 3D modelling using MYSTIC.

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





| Scenario | Land reflection | Ocean reflection | Aerosol properties |
|---|---|---|---|
| Standard | Lambertian albedo 0.05 for λ=450nm 0.3 for λ=850 nm | CM, wind speed 5 m/s | α=1.4 β=0.05 |
| Land | RPV BRDF parameters (ρ0,k,Θ): (0.1,0.78,-0.1) for λ=450 nm (0.2,0.784,-0.083) for λ=850 nm | | |
| Ocean | | Lambertian albedo 0.05 for all wavelengths (CM, wind speed 30 m/s) | |
| Aerosol | | | α=1.4 β (am, pm) = (0.045, 0.055), (0.055, 0.045) |

Table 1. Model input parameter description for the three scenarios regarding land and ocean BRDF and aerosols.





**Figure captions**

Fig.1 Geographic locations of the measurement sites near Grottammare on the Adriatic coast, Italy. Within 20 km from the coastal site, the landscape consists of hills of less than 500 m elevation with patches of forest and agricultural land. The measurements were made in September 2015.

5 The right panel shows the set up for the modeled radiances. The physical scenario is approximated by a 400km x 400km box with a straight coastline at 20° from the north-south direction and a flat topography with 100m elevation. The ocean BRDF is parameterized according to Cox and Munk (CM) with 5 m/s wind speed. Lambertian albedo (0.05, 0.23 and 0.3 for 450, 650 and 850 nm respectively) is assumed for the land reflectivity. The black blobs show the solar azimuths for different solar zenith angles (white numeric labels).

Fig. 2 Measured and modeled radiances at 850 nm at the coastal site at four SZAs in the morning (am) and afternoon (pm). Units are relative units (counts for measurements, transmittance for model simulation) but are irrelevant for the analysis here. The right panel shows the solar azimuths (dark blobs) and the viewing azimuth angles (white dashes) in relation to the coast line (land is green, ocean is blue). Note how the radiance is higher 15 for viewing directions towards the land, as compared to the ocean.

Fig. 3 Asymmetry of the radiance with respect to the principal plane: right-left ratios of the radiances at 70° viewing zenith angle at three wavelengths and four SZAs at the coastal site. The azimuth is relative to the principal plane (dashed lines) which is illustrated in the sketches on the right. The plots have been rotated so that 20 the principal plane is always vertical which facilitates the visualization of the symmetry. At 67.5° SZA pm, the principal plane is perpendicular to the coast and the radiance distribution is symmetric with a ratio of one. Towards noon, the ratios for the longer wavelengths increase as the principal plane aligns with the coastline and maximizes the asymmetry with respect to the high albedo land on the right and the low albedo ocean on the left.

25 Fig. 4 Asymmetry between morning and afternoon: measured and modeled am-pm ratios of the radiances at 70° zenith angle at three wavelengths and four SZAs at the coastal site. The viewing azimuth angles and the coastline are again illustrated on the right. The ratios are higher than one for viewing azimuths where the underlying albedo differs from am to pm.

30 Fig. 5 Investigating the impact on the right-left ratios (am) and am-pm ratios at 850 nm for three model scenarios: modified land and ocean surface reflection models and AOD uncertainty. The measurements are shown as dark blobs for comparison. A difference of beta of 0.01 am to pm changes the radiance and hence the am-pm ratios by up to 10% (indicated as the grey band) which may explain the discrepancy of the standard model scenario with the measurements. The right-left ratios are not affected by AOD uncertainty and are 35 therefore a good indicator that the pasture land BRDF is less appropriate here. The ocean BRDF only has an effect at high SZA.

Fig. 6 Measured and modeled am-pm ratios at the coast for 450 nm and 650 nm as a function of SZA. AOD variations during the day cause the measured ratios to fluctuate. The grey band indicates the maximum 40 variability of the ratio due to an AOD difference of 0.01 (Angstrom beta) between morning and afternoon for



650 nm. Note the splitting of the ratios between the two wavelengths at around 65° SZA, characteristic for the ocean's sun glint at high SZA.

Fig. 7 Measured and modeled am-pm ratios of the zenith radiance at the two locations over the day. Measured
5   ratios are only available in the SZA range 65° - 75° because of clouds around noon. The am-pm ratios at the coast increase with SZA and wavelength with little variation 15 km inland. Right panel: Modeled ratios of zenith radiances at 70° SZA for 450 nm and 650 nm, along the transect perpendicular to the coastline. The am-pm ratio reaches a maximum of about 1.07 directly at the coast and decreases below 1.01 further than 30 km from the coast.





Figures

Figure 1

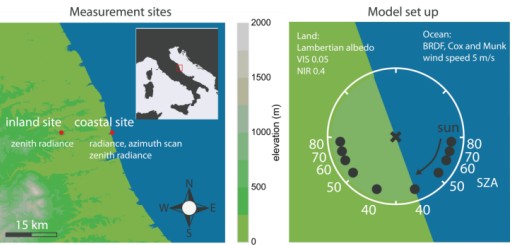

Figure 2

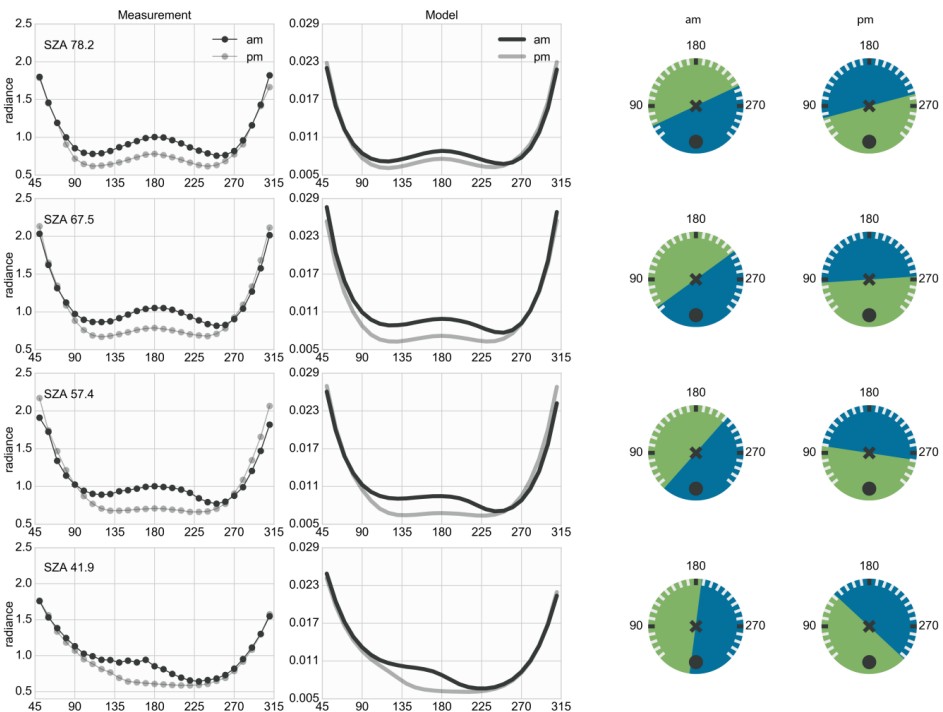





**Figure 3**

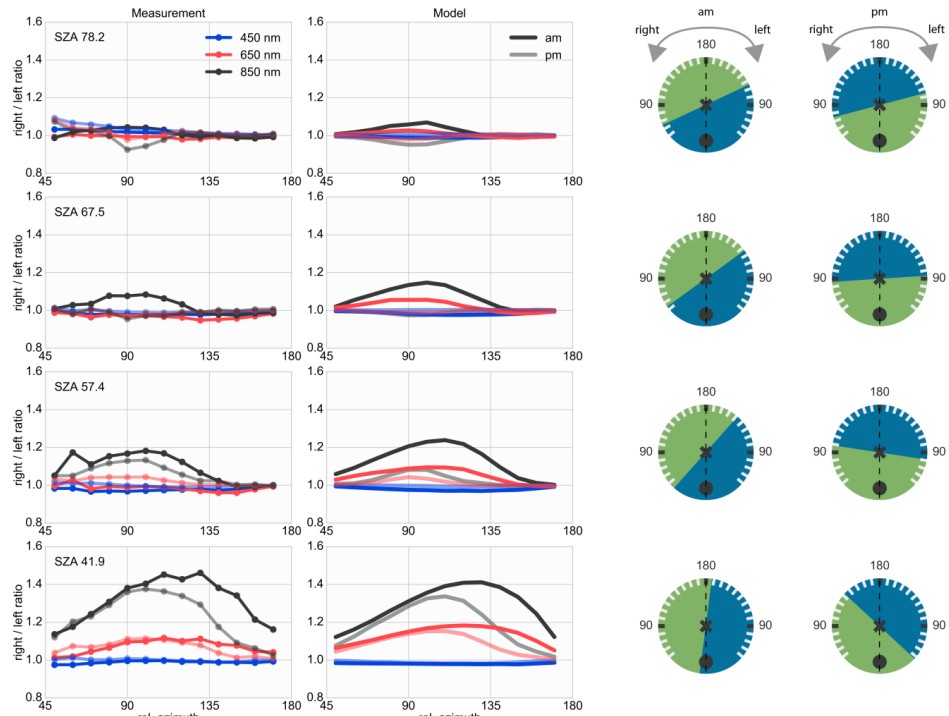





**Figure 4**

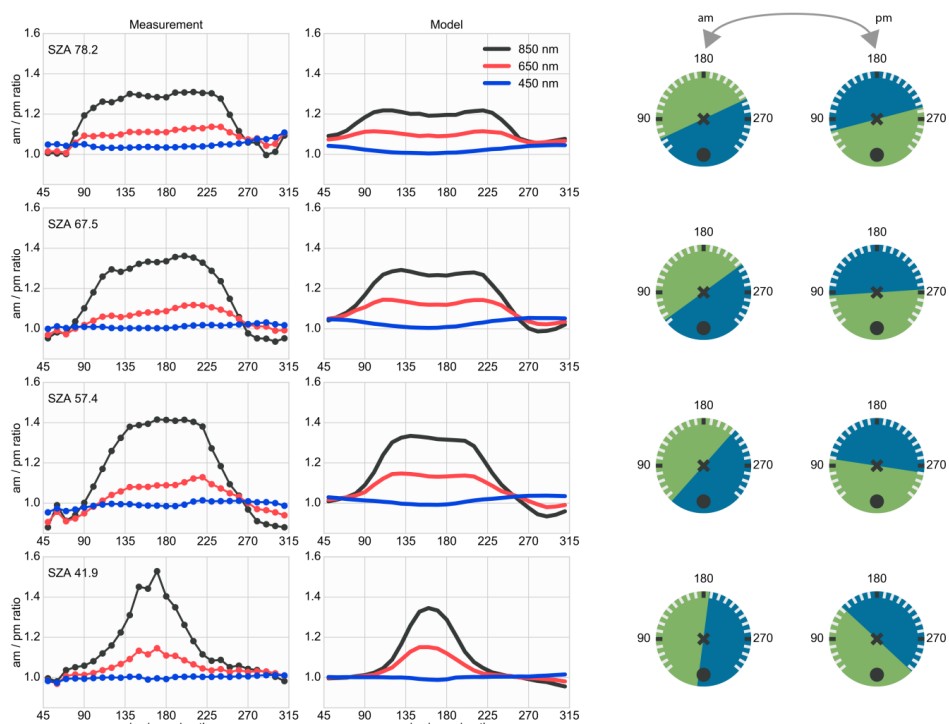





**Figure 5**

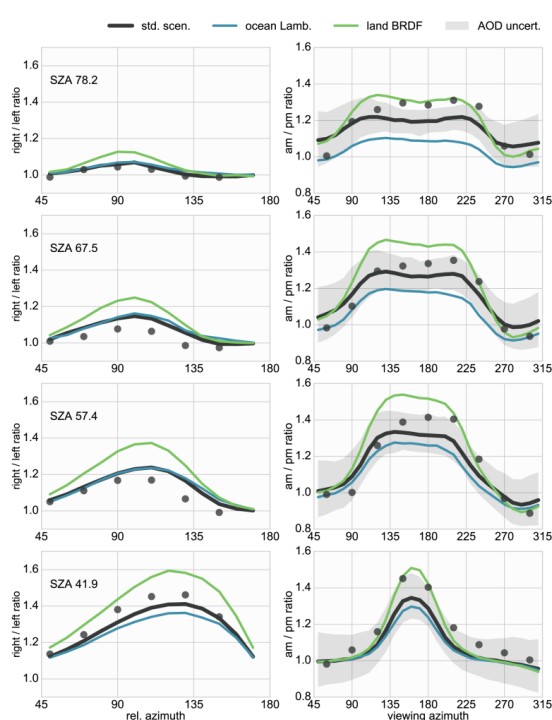





**Figure 6**

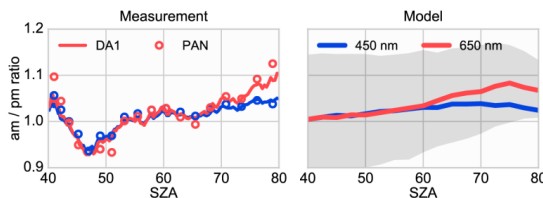

**Figure 7**

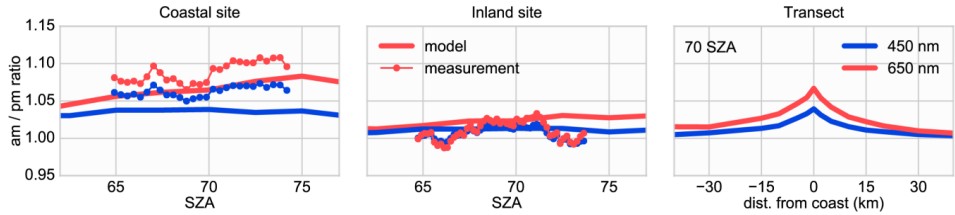

