# Peer review of "Sky radiance at a coastline and effects of land and ocean reflectivities"

_Atmospheric Chemistry and Physics, 2017_

## Referee Comment (RC1) · Anonymous Referee #2 · 17 Sep 2017

The paper "Sky radiance at a coastline and effects of land and ocean reflectivities" by Axel Kreuter et al. describes the impact of heterogeneous albedo environment on sky radiance measurements at a coastal site. The authors analyze the effect of the albedo change between the land and the sea, as well as the effect of the anisotropy of the reflectance of the surfaces. This effect is studied on the almucantar as well as on the zenith radiance.

Although the experimental data is scarce for a full evaluation of the heterogeneous albedo effect, the agreement with the model is correct and the differences obtained with respect to a homogeneous model are very significant. The paper is clearly structured and well written. There are only minor comments.

Page 2, Line 4. From Coakley (2003): "Even though isotropic reflection is not common,

isotropic reflection is often adopted as an approximation because it simplifies estimates of the reflected intensities". So, from the referenced paper it can not be understood that land reflectance is typically diffuse. It is just the simplest approximation, and thus it is typically used.

Page 3, Line 14. Can you provide the zenith radiance error due to be approximated by a $60°$ field of view measurement?

Page 3, Line 23. To my knowledge, the Pandora-2s aerosol product is still in development phase. Can you clarify if AOD measurements are from Pandora or from the auxiliary sun photometer? Can you provide any further information about the auxiliary sun photometer?

Page 4, Line 22. Please clarify why aerosol properties are specified according to the OPAC aerosol type continental average instead of the measured AOD.

Page 7, Line 2. In order to properly establish the variation of the optical thickness, can you please provide mean value and standard deviation of Angstrom exponent along with the beta values?

Fig.2 Even if it is clarified in the caption, magnitude and units should be in the y-axis of the graph.

---

## Referee Comment (RC2) · Anonymous Referee #1 · 18 Sep 2017

The present study deals with spectral sky radiance distribution above a coastline. Measurements of sky radiance distribution are shown and compared with simulations from the Monte Carlo model MYSTIC. In my opinion there is still some need of research in this domain. I agree with the statement of the authors that the findings may be relevant for ground based remote sensing methods among others for the determination of aerosol characteristics. The manuscript is well structured, the methods are sound and include state of the art instrumentation and quality control methods. The 3-D model is a well known high quality radiative transfer model. Conclusion and analysis are reproducible and comprehensible. Since the study includes several innovative aspects and meets the expected level of quality I suggest the acceptance of the manuscript in ACP.

I have following remarks that should - before acceptance - be taken into account.

[Figure]

a) 2. Methods: Concerning the accuracy of the radiance measurements some estimation concerning the measurement uncertainty should be given. It should also be mentioned what would be the discrepancy between measurement and modelling if absolute values were compared. The reason for the uncertainties and why you are only comparing relative measured and modelled values should also be mentioned.

b) You should mention problems related to polarisation and why the polarisation of the sky radiance does not influence the spectrometer measurements

c) In the y axis of fig.2 you should add [rel. unit]

d) 4. conclusions The last section about the relevance of the present research should be extended. What do you think are the next interesting questions? Would it be possible by using absolute values (assuming a better measurement accuracy) to obtain more information about ground albedo?

e) If available, you should at the beginning of the results section add a figure showing the radiance distribution over the whole hemisphere at one wavelength.

---

## Referee Comment (RC3) · Anonymous Referee #3 · 19 Sep 2017

The Review of the paper "Manuscript ID: ACPD-2017-622 Title: Sky radiance at a coastline and effects of land and ocean reflectivities Authors: Axel Kreiter et al.,

The article describes spectral radiance measurements with a 3D-model comparison to analyse the effect of a heterogeneous albedo environment over them. This is the case of sites in the coast where sea water and land surface reflectivities impact over solar irradiance and sky radiance measurements. The authors analyse the different contribution of these surface reflectances and the subsequent influence over the morning/afternoon and left/right radiance data of the coastline site (radiance measured at 70 viewing zenith angle (almucantar data) as well zenith radiances). Six paragraphs are developed in the Results section which also includes a paragraph (3.4) to analyse the model sensitivity to BRDFs water and land changes and aerosol variability. The other

two paragraphs (3.5 and 3.6) are dedicated to zenith radiances. The radiance measurements at 70 degrees or almucantar measurements are carried out with Pandora spectroradiometer and the zenith radiances with the DA1 and DA2 spectroradiometers. It is not clear in section 2 that DA1 and DA2 cannot measure almucantar data and in reality they measure diffuse irradiance. What is the field of view (FOV) of DA1 and DA2 when they measure the zenith radiance? Is this FOV 60 degrees? Clarify better these points in section 2, only at the end (Page 9) this is clear.

Otherwise most of the data are carried out for SZA larger than 60 degrees, which are of interest for high latitude sites but of relative interest for, i.e., the Mediterranean area where precisely the experiment was carried out. Certainly these effects have the mayor influence for high SZA but because of the scarce published radiance data values for lower SZA are of interest (in general most of the scientific community working in these topics of solar radiation, remote sensing, etc., is not so much familiar with radiance data). SZA of $41.9°$ is 1.2 hours from local noon, why not to include 0.5 hours from local noon?

In appearance to section 2 describing spectroradiometers I'm waiting to see radiance spectra in the results section but only values at three specific wavelengths are shown. I consider of interest to see and to analyse the changes of the radiance spectra for some of the items analyzed.

Below, only minor considerations

1. It would be recommendable to add a new figure to Figure 1 as a) or c) showing the geometry of the measurements, observer, sun, sky, to help the reader.

2. Although in the text, in the caption of Figure 2 it must be added that sky radiances correspond to $70°$ viewing zenith angle. Furthermore, it should be mentioned in some part of the text that this type of sky radiance measurements are also called almucantar measurements, it would help to some readers not so familiar with radiance data.

3. Page 3, line 39: Clarify what type of sun-photometer was used. If this was a Cimel sun-photometer, a comparison of radiances with this instrument will be required. This comparison will be a good assessment of the uncertainty or error of measurements (because of the differences between instruments) and hence the observed differences between the measured and model data. Otherwise, specify the type/model and the wavelengths/filters of the sun-photometer.

4. Page 7, line 2: Why the authors model the spectral radiance with the alpha-beta turbidity parameters when they only show the data for three specific wavelengths: 450, 650 and 850 nm (not a given spectrum), why they do not use the AOD for these specific wavelengths (this point is related with my previous comment about the wavelengths of the sun-photometer). In this case the error of simulated data will be minor. However, I can imagine that the differences will be absorbed by the grey band uncertainties shown later in Figure 5.

5. Page 9, line 10. I cannot understand what means "ratio modelled with a constant AOD". Refers it to wavelength or time variation?

In general the paper is near ready for acceptance, it is well structured and written but above considerations must be also taken into account
* * *

---

## Author Comment (AC1) · 19 Oct 2017

Reply to Referee #1

First, we appreciate the positive review and, in particular, the detailed comments to improve the manuscript. The replies to the specific comments:

a) The accuracy of the radiance measurements usually is an important issue when comparing absolute radiances with modeled values. Measuring absolute radiances is a lot more challenging regarding radiometric calibration which was not performed for this campaign. Here, we are concerned with the precision (not accuracy) of the relative stability which has been included in elaborating paragraph 4, page 3, see below, under b).

In any case, since we compare ratios for our conclusions, the calibration requirement is relieved here. The primary reason for investigating ratios (besides the above advantage) has been worked out towards the end of section 3.1, i.e. that the albedo distribution at the coastline breaks the symmetry in the sky radiance. A very powerful way to display a break in symmetry is to show the ratio of values that are expected to be equal in case of symmetry. See page 5, lines 38pp "*In the following, we will investigate the respective ratios which highlight this asymmetry of the radiance above the albedo distribution*" .

b) This is a very valid point, since the polarization of the radiance may affect measurements. All three instruments used in this study are fiber coupled to the input optics with optical fibers (length of 10 m) that are not polarization maintaining. The insensitivity to polarization has been confirmed in the laboratory by rotating a polarizer between a light source and the input optics and monitoring the measured signal. Page 3, lines 33pp has been appended accordingly.

*"The instruments were not calibrated in absolute radiometric units, since we will be considering relative ratios, where the absolute calibration is irrelevant. Only relative radiometric instrument stability has to be ensured, by temperature-stabilizing the instruments. Before the field measurements, DA1 and DA2 were operated together at the coastal site to check instrument stability. From this inter-comparison (and earlier campaigns, see Kreuter et al., 2014) we estimate the precision of these two instruments over the course of the day to about 1%. The precision for the PAN instrument is expected to be of the same order. All instruments are fiber coupled to their respective input optics, with optical fibers that are not polarization maintaining which ensures the instruments' insensitivity to the polarization of the sky radiance."*

c) [rel. units] has been added to the y-label of Fig. 2.

d) Absolute radiances (and global and diffuse irradiances) could indeed be used to obtain more information about ground albedo, especially using advanced algorithms such as the General Retrieval of Aerosol and Surface Properties (GRASP) in combination with satellite information. But this direction is a little off-topic with respect to our goals here. In fact, the coastline is a particularly unsuitable setting to retrieve ground albedo.

However, the comment triggered some more thinking about the "next interesting questions". Our study is focused on the effects of the albedo distribution on the sky radiance. As a further

step, it would be interesting to see the effect on the radiance's polarization (degree and angle of polarization). The polarization is often also measured as additional information for the retrieval of aerosol properties. The final paragraph of our conclusions has therefore be extended to mention this idea: "*Since the degree and angle of polarization may also be used in these retrievals, an interesting question for further studies in the future would be about the effect of inhomogeneous ground reflection on the radiance's polarization.*"

e) We only performed the measurements described in section 2.1. We do have RGB images of the whole hemisphere from the all-sky imager which we used to confirm cloud free conditions, but they are not ideal to show the radiance distribution (although the image, or one of the channels is closely related to the radiance). We have modeled radiance distributions, from the extensive model study prior to the campaign in order to estimate the effects that can be expected and to find the ideal measurement geometry.

So yes, the radiance of the whole hemisphere is of course interesting, but showing sky radiance distributions in a new figure in this study is problematic for many reasons: We could only show model data, and showing the model for one wavelength and SZA would somewhat be an arbitrary choice. Showing distributions for all wavelengths and SZAs would by far not justify the space required in relation to the increase of insight with respect to our conclusions.

---

## Author Comment (AC2) · 19 Oct 2017

Reply to Referee #2

First, we appreciate the positive review and, in particular, the detailed comments to improve the manuscript. The replies to the specific comments:

Page 2, line 4. This a valid point, isotropic reflection is indeed only used as a good approximation. The corresponding sentence has been modified "*Typically, as a simple approximation, land reflectance is assumed diffuse…*".

Page 3, line 14. The zenith radiance measured with a 60° FOV is of course a very crude approximation to the "real" zenith radiance and errors in absolute values will be significant, depending on SZA, aerosol loading etc. However, for the reason outlined in the revision below, the 60° FOV is not really used as an approximation and the difference in the ratios due to the FOV is negligible, page9, line 7pp:

"*This effect concerns a wider range of angles around the zenith, which allows the use of 60° FOV zenith measurements (DA1 and DA2) for the ratios investigated below.*"

This is also confirmed in the comparison between the 2.5° FOV zenith radiance of PAN and the 60° FOV zenith radiance of DA1 shown in Fig 6 and stated on page 9, line 7.

Page 3, line 23. That is correct, the aerosol product of the Pandora-2s is still under development. AOD retrievals from the Pandora have been attempted but were ultimately dismissed due to some remaining issues. In this study, the AOD measurements are exclusively from the sun photometer, which is now described in detail on page 4, lines1 pp. Also a confusing referral to AOD results from the Pandora on page 7, line 9 has been eliminated.

"*Third, the AOD measurements from the sun photometer indicate a constant AOD ($\beta$=0.05) with a remaining uncertainty from the standard deviation of about 0.005.*"

Page 4, line 22. The sun photometer only measures the AOD at four wavelengths without further retrievals of aerosol optical properties which have to be specified for the radiative transfer. We used OPAC type continental average as a sensible approximation which is in accordance with the measured Angstrom alpha value of 1.4.

Page 7, line 2. The variation of Angstrom $\alpha$ is 0.06 (1-sigma) standard deviation which mainly affects the AOD at 450 nm. This fact has been added on page 7, lines 28pp:

"(*the standard deviation of $\alpha$ is 0.06 which would further increase the uncertainty of the AOD, predominantly at short wavelengths*)."

Fig.2. [rel. units] has been added to the y-label of Fig. 2.

---

## Author Comment (AC3) · 19 Oct 2017

**Reply to Referee #3**

First, we appreciate the positive review and, in particular, the detailed comments to improve the manuscript. The replies to the specific comments:

a) The 60° FOV measured by DA1 and DA2 has been clarified on page 3, line 16 "*These measurements will be referred to as zenith radiance with a 60° field of view*". The measurement geometries of each instrument are now also explicitly shown in Fig. 1, which should eliminate any remaining ambiguities.

b) Our measurements are shown for SZA larger than 40° (not 60°) which is representative for mid-latitude locations for most of the year.

c) We have chosen a set of the most representative SZAs (in roughly equal in steps) that illustrate the effects. We do not have a measurement exactly 0.5 hours from local noon, but since it would correspond to 39.5° SZA it would not contribute significant additional information to our data at 41.9° SZA.

d) We have indeed measured complete radiance spectra in the UV-VIS-NIR spectral range which carry an abundance of additional interesting information (e.g. trace gas total columns etc.). Our focus in this study lies on the asymmetries in the sky radiance due to the albedo distribution at a coastline. These effects are showcased by three representative wavelengths, optimizing the ratio of data/conclusion.

The measured total data includes more than 60 spectra of zenith radiance and 30 azimuth scans with each scan consisting of 17 spectra (at each azimuth angle) and scans in the principal plane. Each spectrum consists of 2048 pixels (wavelengths). So the amount of data is significant and the challenge is to distill the information and to show the essential part that contributes to the conclusions. As we assure under *Data availability*, all data are available on request to the corresponding author.

1) Figure1 has been modified, to show the measurement geometry of each instrument. The geometries of observer, sun, sky and coastline is depicted in the left panel of Fig.1 (model set up).

2) 70 ° viewing zenith angle has been added to the caption of Fig 2. Mentioning the almucantar is a valuable improvement (we were a little worried that it would be confusing for the aerosol community where often almucantar is used as jargon for solar almucantar). The almucantar has now been referred to on page 3, line 25 *"...measure the spectral radiance for a set of azimuth angles at 70° zenith angle, i.e. along an almucantar"*.

3) The last paragraph of section 2.1 has been clarified, and details about the sun photometer used (PFR-SPM) are given:

"…*a sun photometer (the precision filter radiometer (PFR-SPM) developed by the Physikalisch-Meteorologisches Observatorium (PMOD) in Davos, Switzerland, for the Global Atmospheric Watch Network) was used to measure the aerosol optical depth (AOD) at four wavelength channels, 368 nm, 412 nm, 501 nm and 862 nm.*"

4) The SPM measures at the wavelengths of 368 nm, 412 nm, 501 nm and 862 nm, from which we determine the Angstrom coefficients. The difference of the Angstrom interpolated AOD at our wavelengths (e.g. 450 nm) are minor. So yes, differences should be absorbed in the uncertainty bands.

5) The sentence, page 9, line 21 has been clarified to "*…constant AOD over the course of the day*".